# Stochastic Gradient Descent for Wind Farm Optimization

**Julian Quick[1], Pierre-Elouan Rethore[1], Mads Mølgaard Pedersen[1], Rafael Valotta Rodrigues[1], and Mikkel Friis-Møller[1]**

[1]Technical University of Denmark, Risø National Laboratory for Sustainable Energy, Frederiksborgvej 399, 4000 Roskilde, Denmark

**Correspondence:** Julian Quick (juqu@dtu.dk)

**Abstract.** It is important to optimize wind turbine positions to mitigate potential wake losses. To perform this optimization, atmospheric conditions, such as the inflow speed and direction, are assigned probability distributions according to measured data, which are propagated through engineering wake models to estimate the annual energy production (AEP). This study presents stochastic gradient descent (SGD) for wind farm optimization, which is an approach that estimates the gradient of the AEP using Monte Carlo simulation, allowing for the consideration of an arbitrarily large number of atmospheric conditions. SGD is demonstrated using wind farms with square and circular boundaries, considering cases with 100, 144, 225, and 325 turbines, and the results are compared to a deterministic optimization approach. It is shown that SGD finds a larger optimal AEP in substantially less time than the deterministic counterpart as the number of wind turbines is increased.

## 1 Introduction

Wind farms are groups of wind turbines that harness the power in the atmospheric boundary layer to provide renewable energy. When a wind turbine absorbs energy from the air, the air downstream of the wind turbine has reduced power, which often reduces the power production of downstream turbines. This is known as the wake effect (Sanderse, 2009; Hasager et al., 2013). When a new wind power plant is to be constructed, optimal turbine locations are determined using engineering wind farm models (Samorani, 2013; Ning. et al., 2020; Annoni et al., 2018). Turbine positions are optimized to exploit the benefits of the local wind resource while avoiding energy losses from turbine wakes. In the wind turbine placement problem, atmospheric conditions, such as the inflow speed and direction, are assigned probability distributions according to measured data. By propagating these probability distributions through the engineering wake model, the annual energy production (AEP) can be estimated. The AEP is often computed using rectangular quadrature, dividing the relevant speeds and directions into equal-sized bins, then computing the expected AEP as the product of the power and probability of each bin, added together, then multiplied by the number of hours per year. The cost of wind farm optimization generally increases with the number of atmospheric conditions considered during AEP computation, and

this expense becomes more extreme as more complex wake models (e.g., RANS) are considered. For example, there are some memory limitations when computing AEP gradients using automatic differentiation with very large wind farms. This has given rise to studies seeking convergence of the AEP, proposing methods such as polynomial chaos expansion (Padrón et al., 2019; Murcia et al., 2015) or Bayesian quadrature (King et al., 2020), to avoid discretizing the input distributions into evenly spaced intervals. In this study, we present an approach for wind farm optimization that estimates the gradient of the AEP using Monte Carlo simulation. This does not require the input to be discretized at all, and allows for the consideration of an arbitrarily large number of atmospheric conditions.

Stochastic gradient descent (SGD) is an optimization algorithm commonly used in machine learning when selecting neural network weights (Ketkar, 2017). The algorithm samples the gradient of a stochastic objective, following the mean gradient by a specified distance, then repeating the process, which amounts to optimizing the expected value of the objective. The SGD algorithm is often enhanced to avoid oscillations caused by large changes in the gradient of the objective (Ruder, 2016). This includes methods to reuse previous gradient information (Qian, 1999), dampen oscillations (Riedmiller and Braun, 1993), or incorporate an estimate of the Hessian matrix (Moritz et al., 2016; Byrd et al., 2016; Liu

et al., 2018; Najafabadi et al., 2017). Kingma and Ba (2014) introduced the Adam SGD algorithm, which reuses gradient evaluations and dampens oscillations, and is the basis of the SGD method we propose in this study.

Interestingly, SGD is not often applied to problems with nonlinear constraints, although it can be fruitful to include nonlinear constraints in the context of training a machine learning algorithm. For example, when recognizing three-dimensional pictures of people, it can be useful to impose a constraint that any person's left arm should be close to the same length as their right arm (Márquez-Neila et al., 2017). Many frameworks have been proposed for constrained SGD, including the log-barrier function (Kervadec et al., 2019), penalty functions (Márquez-Neila et al., 2017), blending barrier and penalty functions (Kervadec et al., 2019), and Riemannian geometry (Roy and Harandi, 2017). In this study, we use a penalty term to transform the constrained problem into an unconstrained optimization.

The wind farm layout optimization problem presents a setting where the objective (AEP) can be formulated as being stochastic (e.g., the AEP is derived from a probability density function), while the constraints (e.g., boundaries and minimum turbine spacing) are firmly deterministic. This manuscript explores the potential benefits of formulating the wind farm layout optimization problem in this way. As part of this, the Adam algorithm is extended to optimize a stochastic objective with deterministic constraints. To the best of the authors' knowledge, this exact algorithm has not been published before.

This study benchmarks the performance of the proposed SGD approach when compared to conventional gradient-based optimization within the TOPFARM framework (DTU Wind Energy Systems, 2023b), considering wind farms with different shapes and sizes. We examine the open-source SLSQP algorithm (Kraft, 1988), which is employed in many engineering frameworks (King et al., 2017; Allen et al., 2020; Wu et al., 2020; Zhang et al., 2022; Zilong and Wei, 2022; Kölle et al., 2022; Clark et al., 2022; Simley et al., 2023) and has been used in previous comparisons of optimization algorithms (Lam et al., 2018; Li and Zhang, 2021; Fleming et al., 2022). The TOPFARM framework has been used with SLSQP in several wind farm optimization studies (Riva et al., 2020; Ciavarra et al., 2022; Criado Risco et al., 2023; Rodrigues et al., 2023). In future work, this approach can be extended to co-optimize layout and control strategy—the SGD framework can naturally incorporate uncertainty quantification when modeling the potential control strategies for potential layouts [similar to the work in Gebraad et al. (2017); Quick et al. (2020); Howland et al. (2022)].

While there are some wind plant optimization studies that resemble our approach, we are not aware of any studies that have applied SGD to the wind farm optimization problem [although SGD has been applied to other problems in engineering optimization, e.g., De et al. (2020); Sivanantham and Gopalakrishnan (2022)]. Several wind farm optimization studies make use of gradient-based optimization techniques (Herbert-Acero et al., 2014; Guirguis et al., 2016; Graf et al., 2016; Gebraad et al., 2017; Baker et al., 2019; Riva et al., 2020; Stanley et al., 2021; Croonenbroeck and Hennecke, 2021). Feng and Shen (2015) present a random search approach, moving the wind turbines one by one using a greedy algorithm. Some studies have employed neural networks to forecast power production (Godinho and Castro, 2021), estimate local atmospheric conditions (Stengel et al., 2020), suggest control strategies (Najd et al., 2020), or optimize engineering wake models (Zhang et al., 2021; Zhang and Zhao, 2022; Hussain et al., 2022), which all use SGD algorithms to the train parameters of the neural networks.

The remainder of the manuscript is the following. Section 2 outlines the SGD and deterministic optimization approaches used in this study. Section 3 details the wind farm optimization application cases examined. Section 4 discusses the results of these optimization comparisons. Section 5 provides conclusions and future research directions.

## 2  Methods

When deciding where to put wind turbines, the typical strategy is to maximize wind farm AEP while ensuring turbines are within the prospective site and are not spaced too closely together. In this study, we examine square and circular wind farms, where the corresponding optimization problems are posed as

$$\begin{aligned}
\underset{\bm{x},\bm{y}}{\text{maximize}} \quad & \text{AEP}(\bm{x},\bm{y}) \\
\text{subject to} \quad & (x_i - x_j)^2 + (y_i - y_j)^2 \geq (N_D D)^2, \quad \forall i \neq j \\
& x_l \leq x_i \leq x_u \\
& y_l \leq y_i \leq y_u
\end{aligned}$$

$$(1)$$

and

$$\begin{aligned}
\underset{\bm{x},\bm{y}}{\text{maximize}} \quad & \text{AEP}(\bm{x},\bm{y}) \\
\text{subject to} \quad & (x_i - x_j)^2 + (y_i - y_j)^2 \geq (N_D D)^2, \quad \forall i \neq j \\
& \sqrt{x_i^2 + y_i^2} \leq R,
\end{aligned}$$

$$(2)$$

respectively, where $\bm{x}$ and $\bm{y}$ are the turbine horizontal and vertical locations, $x_l$ and $x_u$ are the lower and upper horizontal square wind farm boundaries, $y_l$ and $y_u$ are the lower and upper square wind farm vertical boundaries, $R$ is the radius of the circular wind farm, $D$ is the rotor diameter, and $N_D$ is the minimum allowable spacing between turbines measured in rotor diameters. From this point forward, we will use a single variable to represent the x- and y-locations, $\bm{s} = \{\bm{x},\bm{y}\}$. When optimizing square wind farms, $x_l$, $x_u$, $y_l$, and $y_u$ are constant.

The AEP is defined as

$$\text{AEP}(\boldsymbol{s}) = 8760 \int\limits_{0}^{2\pi} \int\limits_{0}^{\infty} P(\boldsymbol{s}, u_\infty, \theta)\pi(u_\infty, \theta)du_\infty d\theta, \qquad (3)$$

where $P$ is power, $\pi$ is probability, $u_\infty$ is the freestream velocity, and $\theta$ is the freestream direction. The 8,760 factor reflects the number of hours per year, converting from units of power to units of energy.

The AEP is typically estimated through rectangular quadrature, where the freestream velocity and direction are discretized using evenly spaced intervals,

$$AEP(\boldsymbol{s}) \approx 8760 \sum_{d=1}^{D} \sum_{u=1}^{U} P(\boldsymbol{s}, \mathcal{U}_u, \theta_d)\rho(\mathcal{U}_u, \theta_d) \qquad (4)$$

where $\mathcal{U}$ is a vector of evenly spaced wind speeds, $\boldsymbol{\theta}$ is a vector of evenly spaced wind directions, and $\rho(\mathcal{U}_u, \theta_d)$ is a probability mass function.

The AEP can also be estimated through Monte Carlo integration,

$$AEP(\boldsymbol{s}) \approx 8760 \frac{1}{K} \sum_{k=1}^{K} P(\boldsymbol{s}, u_\infty^{(k)}, \theta^{(k)}), \qquad (5)$$

where $u_\infty^{(k)}$ and $\theta^{(k)}$ represent draw k of the probability distribution $\pi(u_\infty, \theta)$.

The associated AEP gradient can also be approximated through Monte Carlo simulation:

$$\frac{d}{d\boldsymbol{s}}AEP \approx 8760 \frac{1}{K} \sum_{k=1}^{K} \frac{d}{d\boldsymbol{s}} P(\boldsymbol{s}, u_\infty^{(k)}, \theta^{(k)}) \qquad (6)$$

## 2.1   Stochastic Gradient Descent

SGD is built upon the steepest descent algorithm. Early SGD algorithms added a moving average term (sometimes referred to as "momentum") to avoid spurious oscillations (Tian et al., 2023). The conventional Adam SGD algorithm uses two moving averages: one of the gradient and one of the squared gradient. The ratio of these moving averages is used to determine the search direction. SGD algorithms are often combined with a learning rate scheduler, where the step size of the gradient descent is gradually decreased, allowing the optimization algorithm to hone in on the best solution. While the Adam algorithm is already designed to dynamically change the step size, including a learning rate scheduler can further improve the performance. The conventional Adam algorithm is designed for unconstrained optimization algorithms. In the following, we extend the algorithm to allow for deterministic constraints, which is a case that is common in mechanical engineering and unusual in the context

of training neural networks. The basic idea is to aggregate the constraints to a penalty term with units that are consistent with the objective. The penalty term is designed so that, initially, the penalty gradients are of similar magnitude to the AEP gradients, and so that the penalty gradients overwhelm the AEP gradients as the optimization continues.

The SGD algorithm is shown in Algorithm 1, where $\boldsymbol{s}_0$

---

**Algorithm 1** TOPFARM Stochastic Gradient Descent Implementation

---

$\boldsymbol{m} \leftarrow \boldsymbol{0}, \boldsymbol{v} \leftarrow \boldsymbol{0}, \boldsymbol{s} \leftarrow \boldsymbol{s}_0$
for $i$ in $[0, 1, 2, \ldots, T-1]$
do
     if early_stopping and $\eta_i/\eta_0 \leq$ threshold:
         $\boldsymbol{j} = \alpha_i \frac{\partial \gamma}{\partial \boldsymbol{s}}$
         if $|\boldsymbol{j}| \equiv 0$:
             break
     else:
         $\boldsymbol{j} = -\frac{8760}{K}\sum_{k=1}^{K} \frac{\partial}{\partial \boldsymbol{s}} P(\boldsymbol{s}, u_\infty^{(k)}, \theta^{(k)}) + \alpha_i \frac{\partial \gamma}{\partial \boldsymbol{s}}$
     $\boldsymbol{m} = \beta_1 \boldsymbol{m} - (1-\beta_1)\boldsymbol{j}$
     $\boldsymbol{v} = \beta_2 \boldsymbol{v} - (1-\beta_2)\boldsymbol{j}^2$
     $\hat{\boldsymbol{m}} = \frac{\boldsymbol{m}}{1-(\beta_1)^i}$
     $\hat{\boldsymbol{v}} = \frac{\boldsymbol{v}}{1-(\beta_2)^i}$
     $\boldsymbol{s} = \boldsymbol{s} - \eta_i \hat{\boldsymbol{m}}/\sqrt{\hat{\boldsymbol{v}}}$
     $\eta_i = \mathcal{S}(\eta_0, \delta, i)$
     $\alpha_i = \alpha_0 \frac{\eta_0}{\eta_i}$

---

are the initial turbine positions, $i$ is the iteration number, $\alpha_i$ is referred to as the constraint multiplier, $\gamma(\boldsymbol{s})$ is a penalty function, $P(\boldsymbol{s}, u_\infty^{(k)}, \theta^{(k)})$ is the wind farm power associated with the inflow speed and direction, $u_\infty^{(k)}$ and $\theta^{(k)}$, $K$ is the number of samples employed in each SGD iteration, $\beta_1$ and $\beta_2$ are constants, $T$ is the number of SGD iterations, $\mathcal{S}$ is the learning rate scheduler, and $\eta_i$ is the learning rate. By default, the early stopping option is false.

The spacing between turbines is enforced using a penalty term,

$$\gamma_s = \sum_{\forall i, j > i} \min\left[(x_i - x_j)^2 + (y_i - y_j)^2 - (N_D D)^2, 0\right]. \qquad (7)$$

Similarly, the distance outside of boundaries is enforced using a penalty term. When considering square wind farms this penalty term is defined as

$$\gamma_b = \sum_{i=1}^{N_t} \Big[ \max(x_i - x_{ub}, 0)^2 + \max(x_{lb} - x_i, 0)^2 + \\ \max(y_i - y_{ub}, 0)^2 + \max(y_{lb} - y_i, 0)^2 \Big],$$

and, in the case of circular boundaries, it is defined as

$$\gamma_b = \sum_{i=1}^{N_t} \max\left(\sqrt{x_i^2 + y_i^2} - R, 0\right)^2, \tag{8}$$

where $N_t$ is the number of wind turbines.

The total penalty, $\gamma$, is defined as the sum of these two penalty terms,

$$\gamma(\boldsymbol{s}) = \gamma_s(\boldsymbol{s}) + \gamma_b(\boldsymbol{s}). \tag{9}$$

The gradient of the penalty term, $\gamma$, is scaled before being added to the negative gradient of the AEP using the scaling factor, $\alpha_i$.

In Algorithm 1, the learning rate ($\eta_i$), constraint multiplier ($\alpha_i$), number of SGD iterations ($T$), and the samples per SGD iteration ($K$) are all free parameters. These parameters can be optimized to perform well for individual wind farm optimization problems. But there is no guarantee that these particular parameters will perform well for other wind farm problems—and this meta-optimization can be expensive. In the machine learning community, these parameters are sometimes optimized using evolutionary, grid search, or Bayesian optimization approaches (Alibrahim and Ludwig, 2021). In addition, it is common to schedule the learning rate to decay as the optimization proceeds (You et al., 2019; Denkowski and Neubig, 2017).

We propose a method for setting free parameters to ensure that all units are consistent. The only free parameters we manually set are the number of optimization iterations and the number of power samples per iteration. The optimization generally becomes more accurate and more expensive as these parameters increase, and users are free to balance this tradeoff as they see fit. Our formulation does not guarantee that all intermediate solutions satisfy the constraints, especially in the beginning of the optimization. The constraint multiplier begins on a comparable scale to the AEP, and is scheduled to increase so that the constraint gradients overwhelm the AEP gradients as the optimization progresses. The number of iterations, $T$, can be based on a prescribed computational budget.

We initially attempted this approach using the widely used default parameter values in the original Adam algorithm, $\beta_1 = 0.9$ and $\beta_2 = 0.999$. The parameters can be thought of as adding momentum to the moving averages of the gradient and squared gradient, $\boldsymbol{m}$ and $\boldsymbol{v}$. We found that these default values gave too much emphasis to gradients from the penalty function, launching the turbines away from the boundaries in a dramatic fashion. Instead, we suggest the parameters $\beta_1 = 0.1$ and $\beta_2 = 0.2$, which encode a shorter memory of the presence of the penalty. With these new default parameters, and the learning rate defined below, we observed successful convergence for a wide variety of test cases.

The learning rate, $\eta_i$, can be interpreted as converting $\hat{\boldsymbol{m}}/\sqrt{\hat{\boldsymbol{v}}}$ (with unity units) to distance (units of m). In this study, the learning rate is scheduled to decay according to

$$\begin{aligned} S(t=0) &= \eta_0 \\ S(t=T-1) &= \eta_T \end{aligned}, \tag{10}$$

where $T$ is the number of optimization iterations, $\eta_0$ is the initial learning rate, and $\eta_T$ is the scheduled final learning rate. This final learning rate can be thought of as a solution tolerance for the design variables. In this study, we set $\eta_T = 0.1$m.

The initial learning rate, $\eta_0$, is based on a length scale parameter, $L$, which corresponds to a reasonable initial step size for the optimization. By setting the initial learning rate according to

$$\eta_0 = L = D/5, \tag{11}$$

where $D$ is the turbine rotor diameter, we encourage the turbines to move at most $L$ every optimization iteration.

The learning rate is scheduled to decay as

$$S(\eta_0, \delta, t) = \eta_0 \prod_{i=0}^{t} \frac{1}{1 + i\delta}, \tag{12}$$

where $\delta$ is a parameter that controls the learning rate length, such that the final learning rate is $\eta_T$. The parameter $\delta$ is numerically set as

$$\delta(\eta_0, \eta_T, T) = \underset{\delta}{\mathrm{argmin}} |\eta_T - S(\eta_0, \delta, T)|. \tag{13}$$

The constraint multiplier, $\alpha_i$, can be interpreted as converting the gradient of constrained square distances (in units of m) to AEP gradients. The initial constraint multiplier, $\alpha_0$, is set as the mean absolute AEP gradient divided by the length scale, $L$, so that the separation constraint has a similar scale to AEP gradients,

$$\alpha_0 = \frac{\mathrm{mean}[|\nabla AEP(\boldsymbol{s}_0)|]}{L}, \tag{14}$$

where $\mathrm{mean}[|\nabla AEP(\boldsymbol{s}_0)|$ is the mean of the absolute AEP gradient of the initial guess with respect to each component of the gradient. During each iteration, the constraint multiplier, $\alpha_i$, is scheduled to increase based on the inverse of the learning rate,

$$\alpha_i = \alpha_0 \frac{\eta_0}{\eta_i}. \tag{15}$$

The wind rose samples, $(u_\infty^{(i)}, \theta^{(i)}) \sim \pi(u_\infty, \theta)$, are randomly selected based on the direction frequency and direction-specific Weibull shape and scale parameters. Note that the tilde ($\sim$) denotes a shared probability distribution. After a direction is sampled, the wind speed is sampled as a continuous weibull distributed random variable,

$$u_\infty(\theta) \sim \mathcal{W}[u_\infty, A(\theta), k(\theta)]\,, \tag{16}$$

where the probability density of the Weibull distribution, $\mathcal{W}$, is given by

$$\mathcal{W}(u_\infty, A, k) = \frac{k}{A}\left(\frac{u_\infty}{A}\right)^{k-1}\exp\left[-\left(\frac{u_\infty}{A}\right)^k\right]. \tag{17}$$

## 2.2 Deterministic Approach

The SLSQP algorithm (Kraft, 1988) is selected to be the deterministic optimization algorithm to act as a benchmark to the SGD approach. SLSQP is a conventional deterministic optimization approach. It is employed in many open-source engineering design codes (Allen et al., 2020; Wu et al., 2020; Ciavarra et al., 2022) and has been used in other previous comparisons of optimization algorithms (Lam et al., 2018; Li and Zhang, 2021; Fleming et al., 2022).

The spacing and boundary constraints are passed to the optimizer as individual inequality constraints. The spacing constraints are defined as

$$C_{ij} = (x_i - x_j)^2 + (y_i - y_j)^2 - (N_D D)^2 \quad \forall i, j > i, \tag{18}$$

where $\boldsymbol{C}$ is an upper triangular matrix of nonlinear inequality constraints.

Square wind farm boundaries are represented using four inequality constraints per turbine,

$$\begin{aligned} D_{i1} &= x_i - x_u \\ D_{i2} &= x_l - x_i \\ D_{i3} &= y_i - y_u \\ D_{i4} &= y_l - y_i\,, \end{aligned} \tag{19}$$

and the circular wind farm boundaries are represented with one inequality constraint per turbine,

$$D_i = \sqrt{x^2 + y^2} - R\,, \tag{20}$$

where $\boldsymbol{D}$ is a matrix of boundary constraints that must be less than or equal to zero.

## 3 Application

We apply the optimization approaches discussed above to optimize wind power plants of various sizes using the TOPFARM framework (DTU Wind Energy Systems, 2023b). Each farm consists of turbines with 70-meter hub heights, 80-meter rotor diameters, and 2-megawatt rated powers. Power is computed using PyWake (Pedersen et al., 2019), which is an open-source wake modeling tool that has been used in several related studies (Riva et al., 2020; Rodrigues et al., 2022;

Ciavarra et al., 2022; van der Laan et al., 2022; Fischereit et al., 2022). Power gradients are computed directly from PyWake using automatic differentiation. The power of each turbine is estimated by a combination of velocity deficits predicted by the Bastankhah Gaussian wake model (Bastankhah and Porté-Agel, 2014) using the default parameters in the PyWake tool and the squared sum superposition (Pedersen et al., 2019; DTU Wind Energy Systems, 2023a). We require each turbine to be spaced at minimum two rotor diameters apart ($N_D = 2$). This is imposed as an optimization constraint. We considered wind farms with square and circular boundaries. The square wind farm boundaries are determined as

$$\begin{aligned} x_l &= 0 \\ y_l &= 0 \\ x_u &= D(\sqrt{N_t}-1)\Delta \\ y_u &= D(\sqrt{N_t}-1)\Delta\,, \end{aligned} \tag{21}$$

and, in cases with circular wind farm boundaries, the radius is determined as

$$R = D(\sqrt{N_t}-1)\Delta\,, \tag{22}$$

where the $\Delta$ parameter control the average spacing of the turbines. In this study, $\Delta = 5$.

We use the pyoptsparse driver (Wu et al., 2020) SLSQP (Kraft, 1988) implementation (Virtanen et al., 2020) in TOPFARM. The optimizer was set to run for 300 maximum iterations with a tolerance of $10^{-1}$. The TOPFARM "expected_cost" parameter is set to 10. The turbine coordinates are normalized from 0 to 1. In each optimization iteration, the AEP, and the corresponding gradient, is computed using rectangular quadrature as described in Equation 4, using 360 wind direction bins and 23 wind speed bins, resulting in 8280 power evaluations.

The wind rose, visualized in Figure 1, is based on PyWake's Lillgrund example site. A probability mass function is assigned to different direction bins. Each direction bin is associated with Weibull scale and shape parameters describing the distributions of wind speeds within the sector. This probability mass is derived from seven months of measured data used in a previous study (Göçmen and Giebel, 2016). Each direction bin is 30 degree wide. The reasoning behind this is similar that behind the IEC 614400 power curve standard (International Electrotechnical Commission, 2005)—it is crucial that the reference data consider a statistically significant amount of data in each bin. This coarse direction discretization results in a faster convergence of the estimated probability mass function than a finer discretization would. The continuous probability density function $\pi(u_\infty, \theta)$ is approximated as $\rho(\theta)\pi(u_\infty|\theta)$, where $\rho$ is the previously mentioned probability mass function, linearly interpolated across one-degree bins, and $\pi(u_\infty|\theta)$ is parameterized by direction-specific Weibull shape and scale parameters that are also lin-

early interpolated from the provided data. With this formulation, the likelihood of different wind directions is provided as a probability mass function, $\rho(\theta)$. This probability mass is used as weights passed to the Numpy "choice" function (Harris et al., 2020), allowing the wind direction to be sampled as a discrete random variable. We note that this formulation could be extended to a fully continuous formulation by drawing the direction samples from the inverse of an empirical cumulative direction density function.

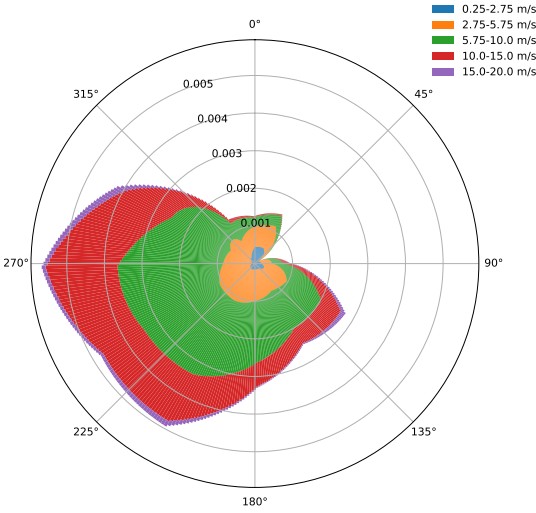

**Figure 1.** Lillgrund wind speed and direction probability mass function with 360 direction bins and five wind speed bins, where the probability mass function is a linear interpolation of coarser measurements.

In all wind farm optimization problems considered, constraint gradients, and the associated penalty function gradients, are computed analytically. The AEP gradient is computed via automatic differentiation. The directions are discretized from 0 to 360 degrees, with one-degree increments. In the deterministic formulation, the discretized wind speed ranges from 3-25 m/s and is divided using increments of 1 m/s.

While each Monte Carlo estimate of AEP has significant error, the average error will be close to zero throughout the course of the SGD optimization. We compare the accuracy of the Monte Carlo approach (Equation 5) and the quadrature approach (Equation 4) to estimate the AEP and the L-2 norm of the AEP gradient. The true values are estimated with a very fine discretization of speed and direction, 0.2 m/s and 0.2 degrees. These are used as reference values to assess the accuracy of the Monte Carlo and quadrature approaches by comparing the errors associated with both approaches as functions of the number of samples and the discretization level, respectively, when analyzing a 100-turbine farm with square boundaries. This convergence analysis is shown in

Figure 2. While some realizations of the Monte Carlo approach yield more accurate results than the quadrature approach, the quadrature approach is generally more accurate than Monte Carlo sampling. The Monte Carlo approach requires on average around ten times as many power evaluations to obtain the same accuracy as the deterministic approach.

In this study, we select 50 samples for every SGD iteration. Figure 3 shows the measured computational cost of computing AEP gradients using the circular wind farm described in this study, with different wind farm sizes. The minimum measured time is reported as the minimum of thirty identical runs on the DTU Sophia supercomputer (Technical University of Denmark, 2019). The computational time generally scales logarithmically with the number of turbines. This is to be expected, as there are more interaction terms in the wake model as more turbines are considered. The computational time does not scale logarithmically with the number of wind rose samples. For small numbers of turbines, evaluating 10 wind rose samples is about as expensive as evaluating 50 samples. This scaling changes as the wind farm grows in size, and it gradually becomes more expensive to sample the wind rose. The evaluation time appears to converge to a logarithmic scaling for large numbers of wind rose samples. These scaling results are likely influenced by memory limitations.

The optimization algorithms are timed based on the time elapsed between the first and final optimization gradient evaluations. Each optimization case is run on first generation AMD EPYC 7351 processors.

## 4 Results and Discussion

In the following subsections, the performance of SGD and SLSQP are compared for wind farms with square and circular boundaries, and the sensitivity of the SGD algorithm is assessed.

### 4.1 Square Wind Farm

The performance of SGD is compared to the deterministic counterpart, considering wind farms with 100, 144, 225, and 324 turbines, with square boundary constraints, using 20 different initial starting conditions to obtain statistically significant results. The AEP, constraint violation ($\gamma$), and time elapsed associated with each optimization solution are plotted in Figure 4. The SGD approach consistently yields higher AEPs than the SLSQP approach when the number of scheduled SGD iterations, $T$, is 2,000. There is a large range of computational times associated with the SLSQP approach, though the computational expense of SLSQP generally grows much larger than SGD as the number of turbines is increased. This is largely due to the nature of the turbine spacing constraint, the size of which grows as the number of turbines squared. SLSQP takes about as much computational time as the SGD approach with 500 scheduled iterations

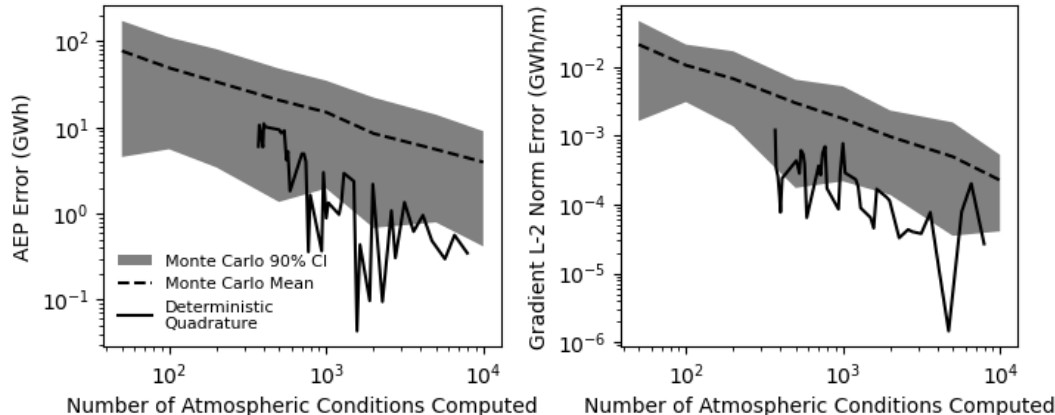

**Figure 2.** Convergence of AEP (left) and L-2 norm of the AEP gradient (right) with respect to the number of samples used in the quadrature and Monte Carlo techniques. The grey cloud shows the 90 percent confidence interval associated with the Monte Carlo approach, using 50 samples. The dashed black line shows the average error associated with the Monte Carlo approach. The solid black line shows the error associated with the deterministic approach.

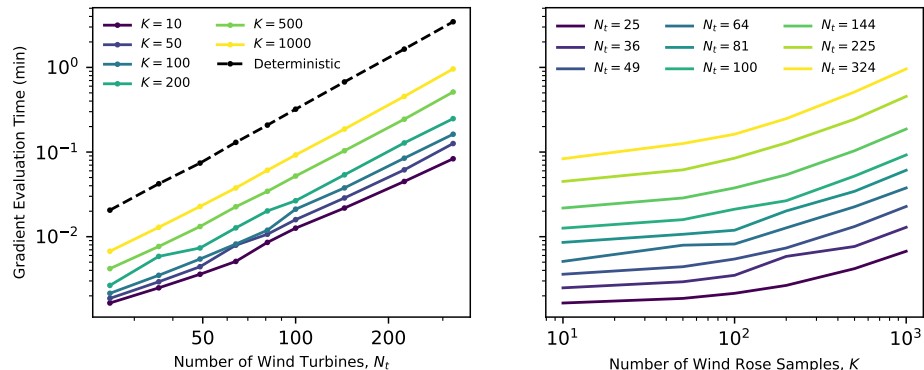

**Figure 3.** Computational time associated with computing the gradient for various wind farm sizes and wind rose sampling strategies. The left panel compares the cost of computing AEP gradients when using different numbers of Monte Carlo samples with the cost of the full factorial wind rose (8,280 samples) for farms with various numbers of wind turbines. The right panel compares the cost of Monte Carlo estimates of the AEP gradient for different numbers of samples of the atmospheric conditions and turbines in the wind farm.

when there are 100 turbines in the square farm. As the number of turbines grows, the average time required by SLSQP becomes more costly than SGD with 2,000 scheduled iterations. The computational cost of SLSQP is a strong function of the initial layout, and the variance of the SLSQP optimization time also increases with the wind farm size. This is due to the complex interaction between the linear boundary constraints and nonlinear spacing constraints. As the proposed SGD formulation does not offer an automatic way to set the number of SGD iterations, $T$, results are shown for different values of $T$. When $T$ is increased, the optimizer finds solutions with larger AEPs, with a computational cost that is approximately proportional to $T$. The SGD solution consistently improves as more optimization iterations are scheduled (larger values of $T$). Results associated with 1,000 SGD iterations tend to yield similar AEP to the SLSQP approach

and results with 2,000 SGD iterations tend yield higher AEPs than the SLSQP designs.

The final layouts associated with one of the random initial conditions used in the 324 turbine analysis, when $T = 2,000$ iterations, is shown in Figure 5. The SGD approach generally identifies solutions with the majority of turbines packed into the side boundaries. The deterministic algorithm also packed turbines into the edges of the farm, although not as many turbines were packed into the East and West boundaries as in the SGD results. The layouts found using the SGD approach tend to have interior turbines that generally appear to be more aligned in the North-South direction than in the deterministic solutions.

The results of the 100-, 144-, 225-, and 324-turbine wind farm optimization cases are summarized in Table 1. The mean time, mean constraint violation, and mean and stan-

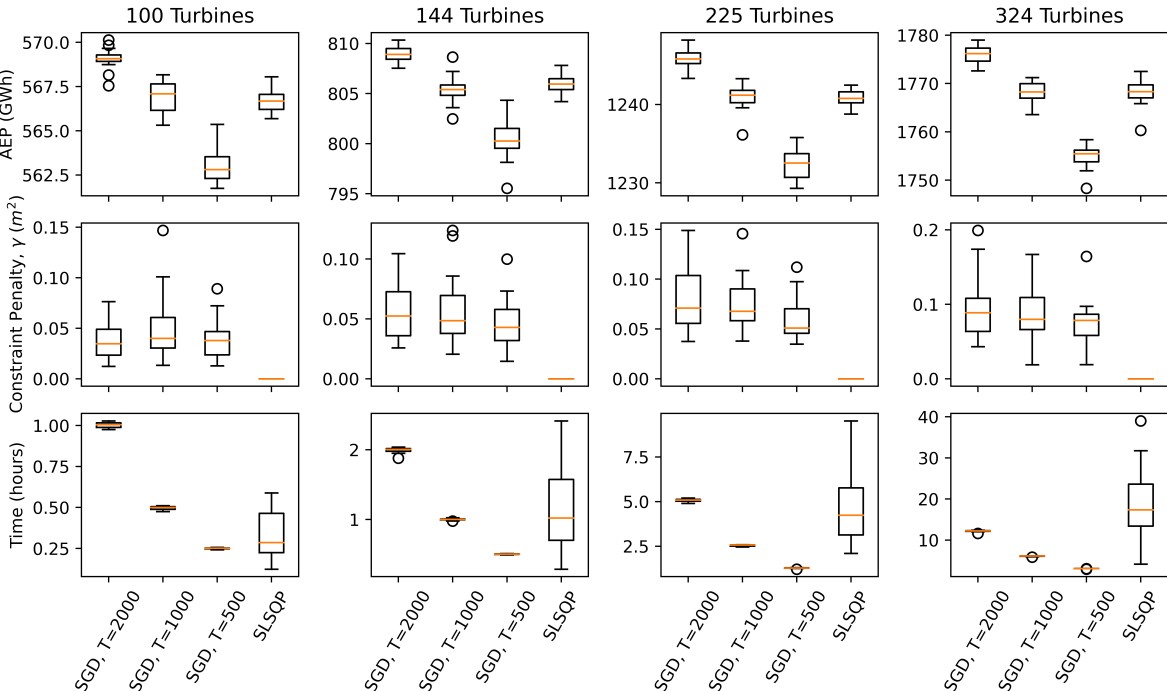

**Figure 4.** Optimization results associated with SGD and SLSQP for square wind farms with 100, 144, 225, and 324 turbines, using 20 random initial starting conditions. The AEP (top panel), constraint penalty (middle panel), and computational time (bottom panel) are plotted as boxplots. The SGD results are plotted for $T = 500, 1{,}000,$ and $2{,}000$ iterations.

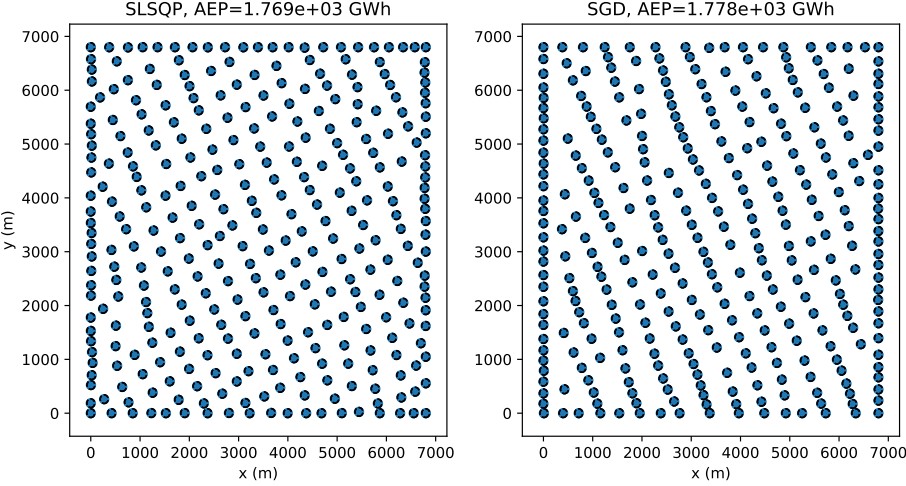

**Figure 5.** The final layouts found using SGD (left) and SLSQP (right) using one of the random initial layouts examined in a 324-turbine wind farm with square boundaries for $T = 2{,}000$ iterations. Each circle a radius of one rotor diameter.

dard deviation of the AEP are reported with respect to the 20 random initial starting conditions. The constraint viola- tion is reported as $\sqrt{\gamma/N_t}$ to quantify the mean length of the constraint violations of each turbine. The final constraint violations can be reduced by lowering the $\eta_T$ parameter. In

all of these cases, the SLSQP optimization resulted in solu- tions with zero constraint violations. This is likely because of the linear formulation of the boundary constraints—when a solution satisfies the spacing constraint, any solutions that satisfy the boundary constraints can quickly be found. SGD

with 2,000 iterations generally yields solutions with AEP that are 0.3-0.5% higher than the solutions found using SLSQP. This is likely because the SGD algorithm is able to better explore the design space by initially relaxing the constraints, allowing for some initial constraint violations.

## 4.2 Circular Wind Farms

To ensure that the previously presented results are not specific to square wind farms, we performed a similar set of analysis examining circular wind farms. The results yield similar trends to the analysis using square wind farms—SGD becomes significantly less time-consuming that SLSQP as the number of turbines increases and generally yields solutions with slightly larger AEPs.

Circular winds farms were optimized using 20 random initial layouts, examining different farm sizes, using the SGD and SLSQP optimization algorithms. The results are summarized in Figure 6. The circular wind farm optimization generally took longer than the square wind farm when using the SLSQP optimizer. This is likely due to the more complicated nature of the circular boundary when using Cartesian coordinates. These results are similar to the results in Section 4.1—as the number of wind turbines and scheduled SGD iterations increases, SGD tends to find solutions with larger AEPs in less computational time.

The results of the circular wind farm optimization are compared between the SGD and SLSQP optimizers in Table 2, where SGD is scheduled to run for 2,000 optimization iterations. SGD generally results in about 0.5% more AEP in significantly less time than SLSQP as the number of turbines is increased. The results area also compared in Figure 7. The SGD optimizer generally results in more turbines on the boundary edge than the SLSQP optimizer.

## 4.3 Sensitivity Analysis

There are several parameters in the SGD algorithm that were tuned to perform reasonably well. In this section, we investigate the sensitivity of the SGD optimization results with respect to the early stopping option in Algorithm 1, the number of Monte Carlo samples per optimization iteration, the learning rate schedule, as well as the initial and final learning rates.

As the optimization progresses, the constraint multiplier, $\alpha_i$, becomes large (approaching 10 as the learning rate approaches 0.1), and the gradients of the AEP are overwhelmed by the gradients of the penalty, which take very little time to compute. This situation can be addressed by using the early stopping option in Algorithm 1. The solution tends to terminate quickly when the optimizer only follows the deterministic gradient (the optimization engine terminates when the constraint gradients are zero). Figure 8 shows the AEP, constraint violation, and computational time associated with five random initial layouts, using threshold parameters of 0.01, 0.05, and 0.1, as well as the SGD algorithm as applied in the previous sections, without the early stopping option activated. The use of each early stopping option results in layouts without constraint violations. As the threshold parameter is increased, the AEP is slightly reduced and the total computational time decreases. A threshold parameter of 0.1 results in approximately 0.3% reduction in AEP and 44% reduction in computation time.

The optimization results presented in this study used 50 power samples per iteration ($K = 50$). We found this to produce high-quality results without incurring unacceptable computational expense. Figure 9 shows the behavior of the SGD approach associated with different different values of $K$, considering 100 turbines with 2,000 scheduled optimization iterations. As $K$ increases, the optimization finds solutions with larger AEPs. There is a small increase in time elapsed and a large increase in the final AEP between the $K = 5$ and $K = 50$ cases, while there is a large increase in time elapsed and a small increase in the final AEP between $K = 50$ and $K = 200$. As $K$ increases, we expect the maximum AEP to reach a plateau and the time and memory required to increase indefinitely. In future work, we plan to explore scheduling $K$ to change as the optimization progresses.

This study used an exotic learning rate scheduler. We tried several schedulers, and observed this one to be the best at finding sufficiently large AEP solutions that reasonably satisfied the imposed constraints. Figure 10 shows the behavior of the SGD algorithm associated with the presented learning rate scheduler, referred to here as the product scheduler; as well as an exponential and a linear decay scheduler. The exponential scheduler quickly diminishes the learning rate, causing the SGD algorithm to become stuck in local minima. The linear transition from large to lower learning rates prevents the SGD algorithm from having sufficient time to follow enlarged constraint gradients. It is possible that the algorithm could be improved by using separate schedulers for the learning rate and constraint multiplier. For instance, it might be more effective to use a linear scheduler to decrease the learning rate and an exponential scheduler to increase the constraint multiplier. We leave this question for future work.

The initial and final learning rates, $\eta_0$ and $\eta_T$, have units of distance and correspond to the initial and final step size of the optimization algorithm. The final learning rate can be interpreted as the degree to which the constraints are to be satisfied, since this will be the step size the optimization algorithm uses when $\alpha_i$ is large and the constraint gradients overwhelm the AEP gradients. This is illustrated in the left panel of Figure 11, which shows the results of several SGD optimizations using different initial and final learning rates. On average, there is a linear relationship between the constraint violation of the solution and $\eta_T$, where the average final constraint violation is approximately two times $\eta_T$. The maximum observed constraint violation is approximately four times $\eta_T$. In addition, there is a trade-off between the AEP

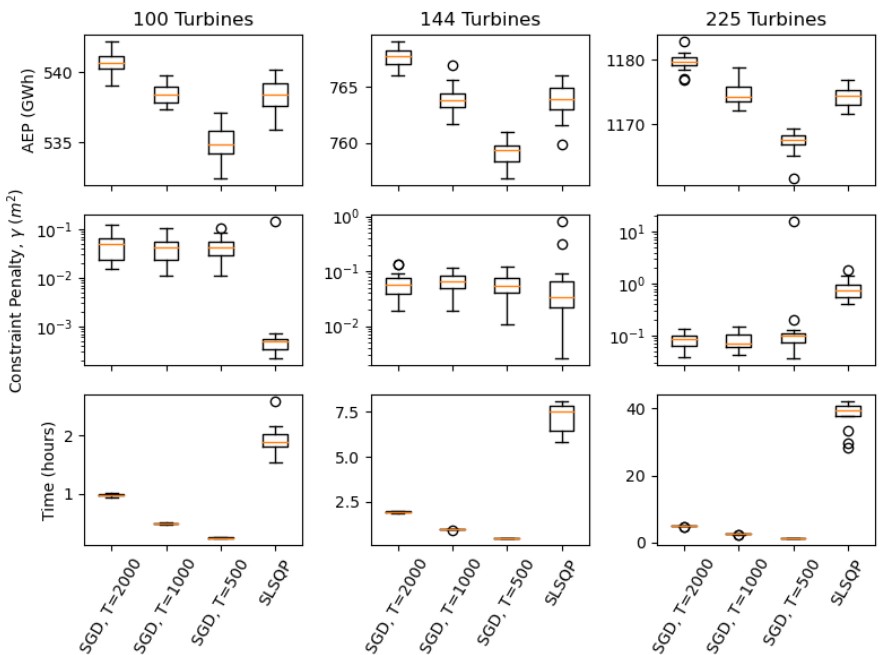

**Figure 6.** Optimization results associated with SGD and SLSQP for wind farms with 100, 144, and 225 turbines, using 20 random initial starting conditions. The AEP (top panel), constraint penalty (middle panel), and computational time (bottom panel) are plotted as box and whisker plots. The SGD results are plotted for $T = 500, 1,000$, and $2,000$ iterations.

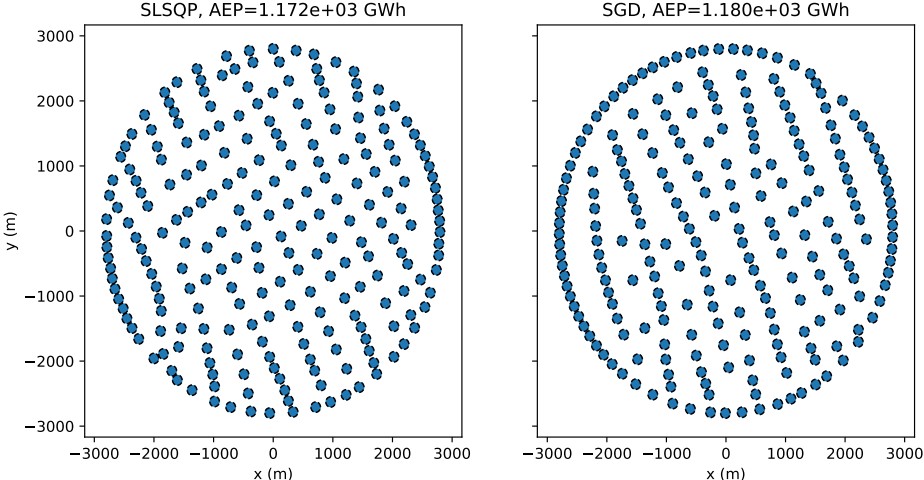

**Figure 7.** The final layouts found using SGD (left) and SLSQP (right) using one of the random initial layouts examined in the 225-turbine wind farm with circular boundaries using $T = 2,000$ iterations. The final turbine layouts are shown as filled circles. Each circle has a radius of one rotor diameter.

and constraint violation of the final solution. This tradeoff is influenced by the initial and final learning rates. It is important to tune the initial learning rate. An initial learning rate that is too low will result in very little exploration. Initial learning rates that are too high will result in a rapid influx of penalty violations that overwhelm AEP gradients through-

out the optimization. From our experiments, we found a step

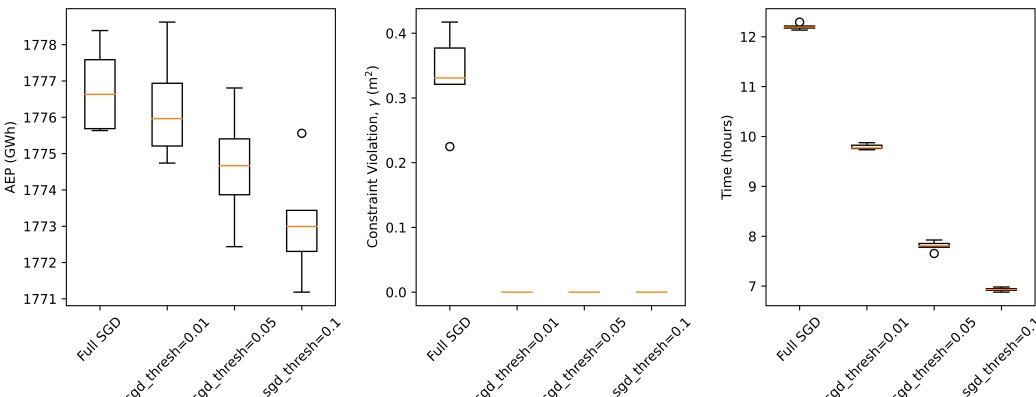

**Figure 8.** Results using different configurations of the "early stopping" option in TOPFARM, as well as the SGD optimization without early stopping, considering the circular wind farm with 225 turbines, with $T = 2,000$, using five random initial layouts.

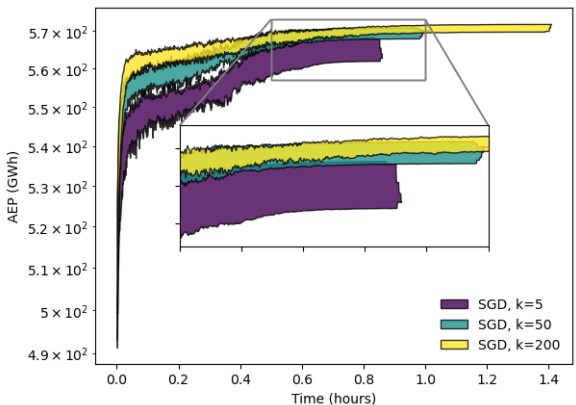

**Figure 9.** Optimization results associated with SGD for a square 100-turbine wind farm using 20 random initial starting conditions and $T = 2,000$. The upper and lower bounds of the results are plotted as a function of the optimization iteration number. The SGD results associated with $K = 5$, 50, and 200 iterations are shown in purple, blue, and yellow, respectively.

size of one fifth of the rotor diameter to produce satisfactory results, as shown in the right panel of Figure 11.

## 5 Conclusions

SGD is a promising optimization tool for wind farm design. Instead of evaluating all anticipated atmospheric conditions during every optimization iteration, SGD randomly samples the defined distributions of atmospheric conditions, resulting in substantially reduced computational time required for each optimization iteration. The total optimization time can be scheduled according to a prescribed computational budget. The presented formulation allows for continuous resolution of uncertain variables, eliminating the need to choose a discretization resolution of atmospheric conditions, such as the wind speed and direction. This technique does not become exponentially more expensive as a greater number of

uncertain parameters is included, allowing for consideration of other atmospheric conditions, such as turbulence intensity, air density, veer, and shear (Saint-Drenan et al., 2020; Duc et al., 2019).

The presented SGD approach was shown to become more effective than a deterministic counterpart as the number of wind turbines increased. SGD yielded slightly higher AEPs than the deterministic approach in substantially reduced computational time. The time required to optimize wind farm layouts can be a major bottleneck in corporate workflows, and the time savings associated with the SGD approach allows engineers to access optimization results sooner than a conventional approach. If the inflow conditions were discretized using extremely small bins, or if several atmospheric conditions were to be considered, we expect that the SGD approach would perform the optimization even faster and more effectively than the deterministic approach.

The SGD approach is a simple framework that is well suited large-scale stochastic wind power plant design optimization challenges. This framework is available in the opensource TOPFARM package. Future work includes: exploring separate schedulers for the constraint multiplier and learning rate and scheduling the number of Monte Carlo samples, $K$, to change as the optimization proceeds.

**Code availability.** The code used in this study is available from DTU Wind Energy Systems' PyWake and TOPFARM repositories (DTU Wind Energy Systems, 2023a, b)

**Author contributions.** JQ and PER designed the experiments. JQ, PER, MMP, and RVR developed the problem formulation. JQ developed the SGD formulation. JQ, RVR, MFM, and MMP performed the simulations. JQ, MMP, and RVP prepared the manuscript with contributions from all co-authors.

**Competing interests.** No competing interests are present

**Acknowledgements.** The authors gratefully acknowledge the computational and data resources provided on the Sophia HPC Cluster at the Technical University of Denmark, DOI: 10.57940/FAFC-6M81

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

**Table 1.** Results of SGD and deterministic optimizations for various square wind farm sizes. Each optimization case is run using 20 random initial starting conditions, and the mean and standard deviation are reported with respect to these 20 initial points. The SGD results are associated with $T = 2,000$ iterations.

| $N_t$ | Case | Mean Time (hours) | Mean AEP (kWh) | AEP Standard Deviation (kWh) | Mean $\sqrt{\gamma/N_t}$ (m) |
|---|---|---|---|---|---|
| 100 | Deterministic | 0.34 | 5.667e+08 | 6.223e+05 | 0.000e+00 |
| | SGD | 1.00 | 5.691e+08 | 5.347e+05 | 1.919e-03 |
| 144 | Deterministic | 1.18 | 8.059e+08 | 9.188e+05 | 0.000e+00 |
| | SGD | 2.00 | 8.089e+08 | 7.727e+05 | 1.642e-03 |
| 225 | Deterministic | 4.79 | 1.241e+09 | 1.004e+06 | 0.000e+00 |
| | SGD | 5.07 | 1.246e+09 | 1.110e+06 | 1.250e-03 |
| 324 | Deterministic | 18.69 | 1.768e+09 | 2.556e+06 | 0.000e+00 |
| | SGD | 12.20 | 1.776e+09 | 1.810e+06 | 9.484e-04 |

**Table 2.** Results of SGD and deterministic optimizations for various circular wind farm sizes. Each optimization case is run using 20 random initial starting conditions, and the mean and standard deviation are reported with respect to these 20 initial points. The SGD results are associated with $T = 2,000$ iterations.

| $N_t$ | Case | Mean Time (hours) | Mean AEP (kWh) | AEP Standard Deviation (kWh) | Mean $\sqrt{\gamma/N_t}$ (m) |
|---|---|---|---|---|---|
| 100 | Deterministic | 1.92 | 5.383e+08 | 1.193e+06 | 2.742e-02 |
| | SGD | 0.98 | 5.407e+08 | 7.844e+05 | 2.203e-03 |
| 144 | Deterministic | 7.25 | 7.638e+08 | 1.512e+06 | 8.648e-02 |
| | SGD | 1.96 | 7.676e+08 | 8.777e+05 | 1.726e-03 |
| 225 | Deterministic | 38.31 | 1.174e+09 | 1.573e+06 | 2.359e-01 |
| | SGD | 4.98 | 1.180e+09 | 1.306e+06 | 1.280e-03 |

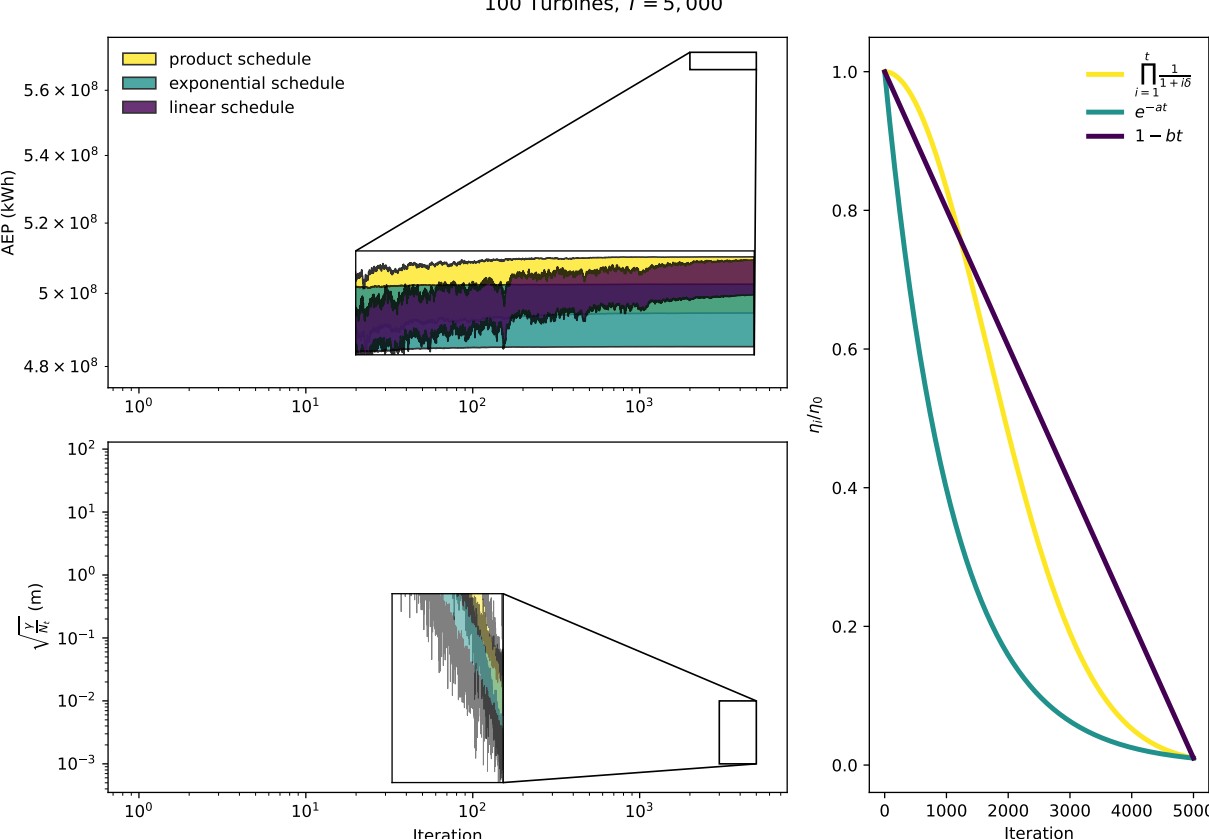

**Figure 10.** Influence of the learning rate scheduler on the SGD optimization, considering a 100-turbine wind farm with a square boundary. The product scheduler is shown in yellow. The exponential scheduler is shown in blue. The linear scheduler is shown in purple. The AEP (upper left panel), constraint penalty function (bottom left panel), and the learning rate decay (right panel) are plotted as a function of the number of optimization iterations. The optimization iteration is denoted as $t$ in the legend of the right panel.

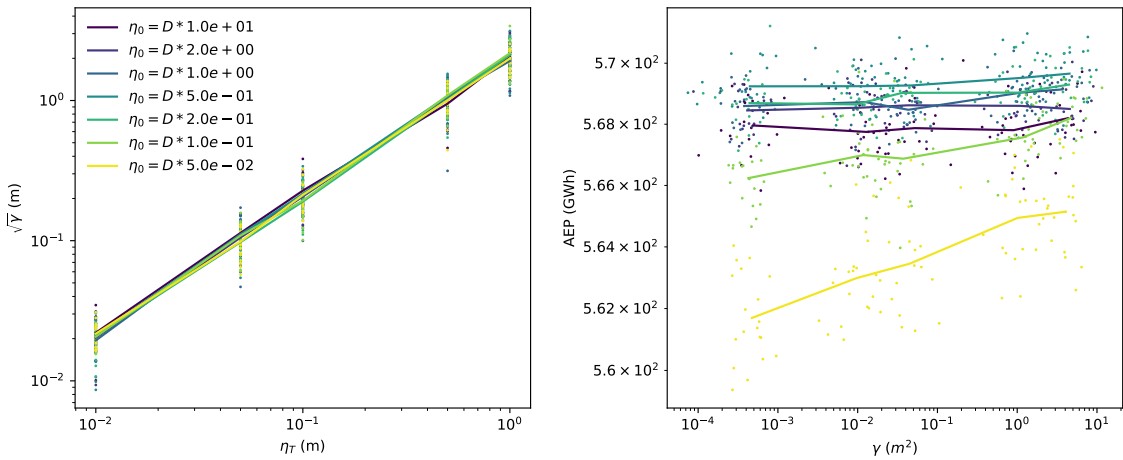

**Figure 11.** (Left) The final constraint penalty plotted against the final learning rate. (Right) The final AEP plotted against the final constraint penalty. The different colors represent different initial learning rates. The results of 20 initial starting positions are plotted as points. The average results of the 20 initial starting conditions are connected as lines. These data are associated with 100-turbine wind farms with square boundaries and $T = 2,000$ iterations.