# Peer review of "Stochastic Gradient Descent for Wind Farm Optimization"

_Wind Energy Science, 2022_

## Referee Comment (RC1)

Review of
*Stochastic Gradient Descent for Wind Farm Optimization*
Julian Quick et al.

Reviewer:

Ahmad Vasel-Be-Hagh
Fluid Mechanics Research Laboratory, Mechanical Engineering Department, Tennessee
Technological University

The article describes using Stochastic gradient descent (SGD) (it is misspelled as "decent" in line 27) to optimize the layout of a wind farm with the objective of maximizing annual energy production (AEP). SGD is a pretty old algorithm; however, the authors used a modified SGC called Adam SGD, developed in 2014.

My primary concern with this article is the novelty. This paper is a single-objective layout optimization that compares the AEP production and computing time for the said algorithm and a second one serving as the baseline. The second algorithm that the article uses as its baseline is Scipy Sequential Least Squares Programming. The two algorithms lead to layouts that generate almost equal AEP (Adam SGD produces ~ 0.2% more, which is not significant). The computational time for the Adam SGD algorithm is nearly 0.5 hours. In comparison, Scipy Sequential Least Squares Programming takes almost 10 hours to identify an optimal layout. This data is for the 100-turbine case, which yields the maximum difference among all presented instances. As the authors also acknowledge, the Scipy Sequential Least Squares Programming is a 30-year-old Fortran code, which one might be able to modify to make it more efficient. Also, other modern algorithms can generate equally efficient layouts as quickly as Adam SGD. Hence, showing that Adam SGD finds layouts that are approximately as efficient as those found by another specific algorithm but does so more quickly does not offer enough novelty, in my opinion. In addition, the industry and the public, including investors, are not concerned with spending several more hours or days finding more efficient algorithms since they invest hundreds of millions of dollars in a wind farm. Speeding up the optimizations is crucial for real-time control of wind farms, not layout optimization, since that is only a one-time job. With that said, again, I must mention that this article does not demonstrate that Adam SGD is the fastest optimizer; it shows that Adam SGD is faster than one other algorithm, leaving the reader to believe there are other algorithms that are faster than Adam SGD.

**Below are my other comments on this article:**

My other concern is about the spacing constraint: "*We require each turbine to be spaced at minimum two rotor diameters apart (ND = 2)*" [Lines 163-164] But then, the article reads: "*All power plants considered in this study have square boundaries so that, in a grid, the turbines would be spaced five rotor diameters apart (Δ = 5).*"
These statements confuse me. Is the spacing 2 or 5 diameters? This matters since ND = 2 is exceptionally tight, and I am not aware of any real wind farm with turbines that are so close. Lillgrund, of which the authors used its wind data for this work, is considered a packed wind farm, in which I believe the two closest turbines in one specific direction are yet apart by more

than 4D. Also, what do authors mean by "in a grid?" Did they define a background grid to discretize the search area or the search area was continuous?

Line 20: *"It can be costly to compute the power associated with each speed and direction combination, as the number of these combinations can be large."* -
I don't believe these computations are costly, and a modern processor can quickly handle such calculations. Also, the wind speed and direction are measured using sensors with limited resolution and other uncertainties, and that's why they are reported in a discretized way using bins.

Lines 23: *"to avoid discretizing the input distributions into evenly spaced intervals. In this study, we present an approach for wind farm optimization that estimates the gradient of the AEP using Monte Carlo simulation. This does not require that the input be discretized at all, and allows for the consideration of an arbitrarily large number of atmospheric conditions".*
Input data are measured in a discretized way. The guide to meteorological instruments and methods of observation recommends measuring wind direction with an accuracy of 5 degrees. Hence, why would feeding the data that is measured discretized in the first place in a "continuous" format into a model help the accuracy?

WMO (World Meteorol. Organ.). 2008. Guide to Meteorological Instruments and Methods of Observations. WMO No 8. Geneva: WMO. 7th ed.

Figure 1: I do not think this figure reflects a realistic representation of Lillgrund's wind conditions. I know that the westerly and southwesterly winds are stronger at this site; however, the data presented in this figure is too extreme. I think this is (at least partly) caused by what I explained in the two previous points. This requires the authors to explain precisely how they converted the data measured discretized to a continuous format. It has to be done by some sort of interpolation. How did they conduct this interpolation? Does the data presented in this figure even sum up to 1 (100%), since the largest frequency shown is about 0.005?

Line 40: *"This term introduces sudden steep gradients in the optimization space, necessitating the use of smaller momentum parameters than are typically employed in the Adam SGD algorithm."*
The article presumes the reader knows the used algorithm; hence, it employs undefined terms. For instance, in the statement above, what are the momentum parameters, and why/how would smaller ones help? This is not defined by this line of the article.

The introduction does not cover the literature on "wind farm layout optimization (hereafter WFLO)." I also find the introduction scattered and distracted with information that does not belong to the introduction, in particular, or this paper, in general. For example, I think the quick review given on lines 59-61 on applications of SGD in areas other than WFLO is distracting. In these applications, SGD tunes a neural network's hyperparameters. Instead, the introduction needs to remain focused on the literature of WFLO, identify gaps in WFLO, and clarify how this new optimization using SGD addresses the identified gaps.

---

## Author Comment (AC1)

**Response to Referee 1**

We greatly appreciate the time taken by the referee to read our manuscript. We have taken into consideration and addressed all comments, questions, and suggestions from the reviewer, and we feel that the revised manuscript is now substantially stronger as a result. Changes made to the text at the request of the reviewer have been highlighted in blue in the revised manuscript. In the following, reviewer comments are repeated in italics and our responses are provided in the bulleted sections of text.

**Major Comments**

*The article describes using Stochastic gradient descent (SGD) (it is misspelled as "decent" in line 27) to optimize the layout of a wind farm with the objective of maximizing annual energy production (AEP).*

- We corrected the typo to read "descent" instead of "decant" and ensured this mistake did not occur elsewhere.

*SGD is a pretty old algorithm; however, the authors used a modified SGC called Adam SGD, developed in 2014.*

- We added to the introduction on P2L43-47 and Methods on P4L96-106 to emphasize that the Adam SGD algorithm is primarily used in the context of unconstrained optimization.

- We are not aware of a similar implementation of the Adam algorithm that allows for deterministic constraints. Therefore, although we feel the primary novelty of this paper is applying SGD to wind farm optimization, we believe that the paper is also novel in that the particular implementation of the Adam algorithm is unique with respect to how the deterministic constraints are considered, which we have now noted on P245-L47 and P4L103-108.

*My primary concern with this article is the novelty. This paper is a single-objective layout optimization that compares the AEP production and computing time for the said algorithm and a second one serving as the baseline.*

- We added to the introduction to highlight that the novelty of the manuscript is the application of SGD to wind farm optimization (P2L43-47).

- We added an introduction to the presented SGD algorithm in the Application on P4L103-108 to explain that our algorithm is an extension of the classic adam algorithm.

*The second algorithm that the article uses as its baseline is Scipy Sequential Least Squares Programming. The two algorithms lead to layouts that generate almost equal AEP (Adam SGD produces 0.2% more, which is not significant). The computational time for the Adam SGD algorithm is nearly 0.5 hours. In comparison, Scipy Sequential Least Squares Programming takes almost 10 hours to identify an optimal layout. This data is for the 100-turbine case, which yields the maximum difference among all presented instances.*

- Since submitting the original manuscript, we have implemented the SGD approach in TOP-FARM. We have updated the results to show the latest results of the SGD/SLSQP analysis, fully implemented within TOPFARM.

- The previous analysis included an SLSQP implementation that computed constraint gradients using finite differences. This amounted to long deterministic optimization times.

- TOPFARM computes the constraint gradients analytically (as is now noted on P9L217-218), significantly reducing the deterministic computational cost, and slightly increasing the SGD computational cost.

*As the authors also acknowledge, the Scipy Sequential Least Squares Programming is a 30-year-old Fortran code, which one might be able to modify to make it more efficient. Also, other modern algorithms can generate equally efficient layouts as quickly as Adam SGD. Hence, showing that Adam SGD finds layouts that are approximately as efficient as those found by another specific algorithm but does so more quickly does not offer enough novelty, in my opinion.*

- Since writing the article, we have incorporated SGD into the TOPFARM framework, allowing for the examination of larger wind farms. Our results indicate that SGD consistently produces about 0.3-0.5% more AEP than the deterministic counterpart. While we believe this is a useful data point, we mainly wish to highlight the difference in computational cost between the two approaches

- We agree that the comparison to SLSQP may have seemed arbitrary in the original text. We modified the introduction (P2L48-57) and Methodology (P7L171-174) to explain that SGD was developed for the TOPFARM framework, and that TOPFARM and other engineering design codes are often run using SLSQP.

- We include references to some studies that have used TOPFARM with SLSQP in the Introduction on P2L48-54 to help give context for this work.

*In addition, the industry and the public, including investors, are not concerned with spending several more hours or days finding more efficient algorithms since they invest hundreds of millions of dollars in a wind farm. Speeding up the optimizations is crucial for real- time control of wind farms, not layout optimization, since that is only a one-time job.*

- Based on our conversations with companies that design wind farms, computational time is a factor when determining the appropriate optimization approach. While developers may not have significant time constraints in developing a layout, turbine manufacturers are often asked to provide suggested layouts, which can amount to a large number of wind farms to optimize.

- In fact, this work was partially funded by a wind energy company interested in speeding up their optimization capabilities. At the extreme, it is obviously impossible to work with an optimization that takes years to complete.

- We note on P6L134-136 and P18L337-338 that our SGD implementation allows for a computational budget to be decided upon before starting the optimization, which can allow for greater organization when planning project design timelines.

*With that said, again, I must mention that this article does not demonstrate that Adam SGD is the fastest optimizer; it shows that Adam SGD is faster than one other algorithm, leaving the reader to believe there are other algorithms that are faster than Adam SGD.*

- We added text to P2L45-46 to attempt to clarify that this manuscript primarily seeks to demonstrate that the SGD algorithm seems particularly well-suited for wind farm optimization.

- We elaborate on this point on P2L43-47, explaining that SGD is distinct in that it naturally allows the problem to be extended to allow for more uncertain variables.

**Minor Comments**

1) *My other concern is about the spacing constraint: "We require each turbine to be spaced at minimum two rotor diameters apart (ND = 2)" [Lines 163-164] But then, the article reads: "All power plants considered in this study have square boundaries so that, in a grid, the turbines would be spaced five rotor diameters apart ($\Delta = 5$)." These statements confuse me. Is the spacing 2 or 5 diameters? This matters since ND = 2 is exceptionally tight, and I am not aware of any real wind farm with turbines that are so close. Lillgrund, of which the authors used its wind data for this work, is considered a packed wind farm, in which I believe the two closest turbines in one specific direction are yet apart by more pg. 2 than 4D. Also, what do authors mean by "in a grid?" Did they define a background grid to discretize the search area or the search area was continuous?*

- We added text in the application P8L193 and P8L98 to better clarify that the minimum rotor diameter ($N_D$) is a constraint imposed on the optimization problem and that the averaging spacing ($\Delta$) was used to initialize the turbine layout and define the boundary locations.

- Thanks to the reviewer's comment, we realized that the $N_t$ was prematurely assigned as 2 in Equation 19 in the original manuscript. We have corrected this in the manuscript, which hope will also enhance clarity regarding the difference between $\Delta$ and $N_D$.

2) *Line 20: "It can be costly to compute the power associated with each speed and direction combination, as the number of these combinations can be large." - I don't believe these computations are costly, and a modern processor can quickly handle such calculations. Also, the wind speed and direction are measured using sensors with limited resolution and other uncertainties, and that's why they are reported in a discretized way using bins*

- We agree that it is somewhat subjective what is "costly" and what is not. To avoid this ambiguity, we modified the text (P1L20-23) to instead state the following.

- The cost of wind farm optimization scales with the number of speeds and directions.

- We have encountered memory limitations when computing gradients in large wind farms.

- As some wind farm models (e.g., RANS) can be time-consuming to run, it can be useful to investigate approaches designed to minimize the number of required flow cases to consider when computing AEP, which examined by the cited studies [1, 2, 3] .

3) *Lines 23: "to avoid discretizing the input distributions into evenly spaced intervals. In this study, we present an approach for wind farm optimization that estimates the gradient of the AEP using Monte Carlo simulation. This does not require that the input be discretized at all, and allows for the consideration of an arbitrarily large number of atmospheric conditions". Input data are measured in a discretized way. The guide to meteorological instruments and methods of observation recommends measuring wind direction with an accuracy of 5 degrees. Hence, why would feeding the data that is measured discretized in the first place in a "continuous" format into a model help the accuracy? [4]*

- We agree that this is a potentially confusing point. To add clarity, we stress that it is important to discretize the atmospheric conditions when measuring the frequency of occurrence, similar to the steps taken to construct a power curve, in the Methodology on P9L206-210.

- We further clarified on P9L204-216 and in the figure caption of Figure 1 that, after the measurements are taken, TOPFARM has the functionality to interpolate these probability

densities, so that the range of potential atmospheric conditions can be fully represented, and that we use a linear interpolation scheme with 1 m/s and 1° bins.

4) *Figure 1: I do not think this figure reflects a realistic representation of Lillgrund's wind conditions. I know that the westerly and southwesterly winds are stronger at this site; however, the data presented in this figure is too extreme. I think this is (at least partly) caused by what I explained in the two previous points. This requires the authors to explain precisely how they converted the data measured discretized to a continuous format. It has to be done by some sort of interpolation. How did they conduct this interpolation? Does the data presented in this figure even sum up to 1 (100%), since the largest frequency shown is about 0.005?*

- The revised version of Figure 1 has 360 direction bins and 5 wind speed bins, so we can expect the probability mass to have an average value of $\frac{1}{360 \times 5} \approx 0.0005$. We affirm that our plotting routine shows a correct probability mass function.

- The revised figure caption explicitly states that this probability mass is obtained via linear interpolation of a probability mass developed using a coarser discretization.

- The original plot considered wind speeds strictly greater than 5 m/s. We revised the plot to include low wind speeds, which we believe now better resembles other published works (e.g., Figure 2(A) in [5]).

- We added a note about the context of this data. We associate this data with Lilligrund because that is how the data is labeled in the PyWake software's selection of example data sets. The data is based on seven months of measurements from the Lilligrund site, as described in Section 2.1 of [6].

5) *Line 40: "This term introduces sudden steep gradients in the optimization space, necessitating the use of smaller momentum parameters than are typically employed in the Adam SGD algorithm." The article presumes the reader knows the used algorithm; hence, it employs undefined terms. For instance, in the statement above, what are the momentum parameters, and why/how would smaller ones help? This is not defined by this line of the article.*

- We agree that this was a potentially confusing point. To avoid this confusion, the sentence in question has been removed from the introduction of the manuscript.

- The concept of momentum is now first introduced in the Methodology in P4L97-98.

6) *The introduction does not cover the literature on "wind farm layout optimization (hereafter WFLO)." I also find the introduction scattered and distracted with information that does not belong to the introduction, in particular, or this paper, in general. For example, I think the quick review given on lines 59-61 on applications of SGD in areas other than WFLO is distracting. In these applications, SGD tunes a neural network's hyperparameters. Instead, the introduction needs to remain focused on the literature of WFLO, identify gaps in WFLO, and clarify how this new optimization using SGD addresses the identified gaps.*

- We added to the introduction to give a better overview of WFLO, including the following references: [7, 8, 9, 10, 11, 12, 13, 14, 15, 16, 15, 17, 18].

- we added material to the introduction on P2L43-47 to more clearly ground our problem and why it addresses an open question:

  The wind farm layout optimization problem presents a setting where the objective (annual energy production) can be formulated as being stochastic (e.g., the AEP is derived from a probability density function), while the constraints (e.g., boundaries and minimum turbine spacing) are firmly deterministic. This manuscript explores the potential benefits of formulating the wind farm layout optimization problem in this way.

- We believe that the revised introduction gives better context to the layout optimization problem, motivates the use of SGD, and makes clear the challenges of SGD with constraints.

**References**

[1] Andrés Santiago Padrón, Jared Thomas, Andrew PJ Stanley, Juan J Alonso, and Andrew Ning. Polynomial chaos to efficiently compute the annual energy production in wind farm layout optimization. *Wind Energy Science*, 4(2):211–231, 2019.

[2] JP Murcia, Pierre-Elouan Réthoré, Anand Natarajan, and John Dalsgaard Sørensen. How many model evaluations are required to predict the aep of a wind power plant? In *Journal of Physics: Conference Series*, volume 625, page 012030. IOP Publishing, 2015.

[3] Ryan King, Andrew Glaws, Gianluca Geraci, and Michael S Eldred. A probabilistic approach to estimating wind farm annual energy production with bayesian quadrature. In *AIAA Scitech 2020 Forum*, page 1951, 2020.

[4] WMO No. 8. guide to meteorological instruments and methods of observation. *World Meteorological Organization*, 2008.

[5] Yulong Ma, Cristina L Archer, and Ahmad Vasel-Be-Hagh. Comparison of individual versus ensemble wind farm parameterizations inclusive of sub-grid wakes for the wrf model. *Wind Energy*, 25(9):1573–1595, 2022.

[6] Tuhfe Göçmen and Gregor Giebel. Estimation of turbulence intensity using rotor effective wind speed in lillgrund and horns rev-i offshore wind farms. *Renewable energy*, 99:524–532, 2016.

[7] Ryan N King, Katherine Dykes, Peter Graf, and Peter E Hamlington. Optimization of wind plant layouts using an adjoint approach. *Wind Energy Science*, 2(1):115–131, 2017.

[8] Ti Zilong and Deng Xiao Wei. Layout optimization of offshore wind farm considering spatially inhomogeneous wave loads. *Applied Energy*, 306:117947, 2022.

[9] Konstanze Kölle, Tuhfe Göçmen, Irene Eguinoa, Leonardo Andrés Alcayaga Román, Maria Aparicio-Sanchez, Ju Feng, Johan Meyers, Vasilis Pettas, and Ishaan Sood. Farmconners market showcase results: wind farm flow control considering electricity prices. *Wind Energy Science*, 7(6):2181–2200, 2022.

[10] Caitlyn E Clark, Garrett Barter, Kelsey Shaler, and Bryony DuPont. Reliability-based layout optimization in offshore wind energy systems. *Wind Energy*, 25(1):125–148, 2022.

[11] Eric Simley, Dev Millstein, Seongeun Jeong, and Paul Fleming. The value of wake steering wind farm control in us energy markets. *Wind Energy Science Discussions*, pages 1–26, 2023.

[12] Nicholas F Baker, Andrew P Stanley, Jared J Thomas, Andrew Ning, and Katherine Dykes. Best practices for wake model and optimization algorithm selection in wind farm layout optimization. In *AIAA Scitech 2019 forum*, page 0540, 2019.

[13] J. Criado Risco, R. Valotta Rodrigues, M. Friis-Møller, J. Quick, M. Mølgaard Pedersen, and P.-E. Réthoré. Gradient-based wind farm layout optimization with inclusion and exclusion zones. *Wind Energy Science Discussions*, 2023:1–24, 2023.

[14] Andrew PJ Stanley, Owen Roberts, Jennifer King, and Christopher J Bay. Objective and algorithm considerations when optimizing the number and placement of turbines in a wind power plant. *Wind Energy Science*, 6(5):1143–1167, 2021.

[15] Peter Graf, Katherine Dykes, George Scott, Jason Fields, Monte Lunacek, Julian Quick, and Pierre-Elouan Rethore. Wind farm turbine type and placement optimization. In *Journal of Physics: Conference Series*, volume 753, page 062004. IOP Publishing, 2016.

[16] David Guirguis, David A Romero, and Cristina H Amon. Toward efficient optimization of wind farm layouts: Utilizing exact gradient information. *Applied energy*, 179:110–123, 2016.

[17] Pieter Gebraad, Jared J Thomas, Andrew Ning, Paul Fleming, and Katherine Dykes. Maximization of the annual energy production of wind power plants by optimization of layout and yaw-based wake control. *Wind Energy*, 20(1):97–107, 2017.

[18] Jeffery Allen, Ryan King, and Garrett Barter. Wind farm simulation and layout optimization in complex terrain. In *Journal of Physics: Conference Series*, volume 1452, page 012066. IOP Publishing, 2020.

---

## Author Comment (AC2)

**Response to Referee 2**

We greatly appreciate the time taken by the referee to read our manuscript. We have taken into consideration and addressed all comments, questions, and suggestions from the reviewer, and we feel that the revised manuscript is now substantially stronger as a result. Changes made to the text at the request of the reviewer have been highlighted in blue in the revised manuscript. In the following, reviewer comments are repeated in italics and our responses are provided in the bulleted sections of text.

**Comments**

1) *The paper is well written and explores an idea worth considering. My main concern is the omission of discussions surrounding accuracy.*

- In an effort to include greater discussions surrounding accuracy, we included material to assess the accuracy of the AEP approximation, as well as the accuracy of the final solution with respect to the imposed constraints. This material is described in the next two bullet points.

- We have included convergence analysis of the Monte Carlo and quadrature approximation in the Application in Figure 2 P10L221-230 in an effort to directly compare the accuracy of the deterministic and stochastic approaches in computing AEP.

- We also included analysis of how well the constraints are satisfied in the Results in Figures 11 and P17L323-P18L333. The left panel of Figure 11 shows the relationship between final scheduled learning rate and the constraint violations of the final solution. The right panel of Figure 11 shows the trade-offs between the AEP and constraint violation of the solutions associated with different initial and final learning rates.

2) *- operating on continuous PDFs isn't so much of an advantage as it is made out to be. And really pretty much any of the UQ methods could be thought of acting on a continuous input (the discretization can be considered part of the method, in the same way that random sampling is part of the method of MC).*

- We have removed references to the benefits of the continuous formulation in the abstract and future work section of the conclusions.

- We still believe that this method will scale favorably as more atmospheric conditions (shear, stability, etc.) are considered, as noted on in the conclusions on P19L330-341.

3) *- the literature review on SGD has good detail, but the lit review is light on other topics that this paper is based on. Namely, the proposed approach is a gradient-based one and there is no discussion of any prior work in this regard, other than the reference to the review paper by Herbert-Accero (which is a somewhat ironic choice as that review paper shows very few gradient-based approaches and is pretty dismissive of the approach in general).*

- We added additional review of work that uses gradients in the introduction on P2L48-P3L62 [1, 2, 3, 4, 5, 6, 7, 8, 9, 10, 11, 12, 13, 14, 15, 16, 17, 18, 19, 20, 21]

- We added references to give a greater perspectives on this rich topic on P1L14 [22, 23].

4) *- Algoirthm 1 could use more introduction. Although I am already familiar with SGD, as noted by the authors it is not in regular use by the wind farm optimization community and so needs more explanation than just an algorithm dump.*

- We agree that this is a good idea. We added an introductory paragraph before Algorithm 1 in the Methodology on P4L97-107 to provide context to the algorithm and explain why our formulation is novel.

5) - *The SGD method notes a mix of AD and finite difference, but in the deterministic approach it is not stated how you compute the derivatives and of course this will affect the time performance.*

- This is an important point to consider. Since submitting the manuscript, we have implemented the SGD algorithm in TOPFARM (which is now noted on P2L49), which also has an SLSQP implementation through the OpenMDAO package.

- We have updated the manuscript to reflect the latest benchmarking results using TOPFARM, where the gradients of the constraints are computed analytically (which is noted on P9L217-218).

- This is in contrast to the original manuscript, where constraints were computed via finite difference. The TOPFARM implementation substantially speeds up the SLSQP algorithm. However, as larger wind farms are examined, the same overall trend emerges – the computational cost of SGD scales more favorably than SLSQP.

6) - *Nowhere is the accuracy of the method shown. Undoubtedly MC can be much faster than a dense rectangule rule, but of course the real question is at what level of accuracy? One would like to see how many random samples are required to achieve the same accuracy.*

- We added analysis and discussion of the accuracy and expense of computing AEP and the associated gradients in the Application in Figure 2 and P10L221-230.

7) - *I realize this is somewhat difficult to directly compare as you are using a stochastic gradient and so allowing for inaccuracy is built into the method, but this is still a topic that one should be more transparent on. Especially since the main comparison is a very dense rectangle rule combined with SLSQP. I don't have a problem with that choice, but lots of things are faster than that combination. It shows promise, but must be careful on concluding anything more than that.*

- We agree that it is best to be as transparent as possible regarding the accuracy of these methods. We also agree that it is difficult to compare accuracy of the methods, as the SGD approach is designed to expect significant errors in the gradient evaluation, using a moving average to find the optimum.

- We included some discussion of the convergence of the AEP using quadrature and Monte Carlo approaches on P10L221-230/Figure 2. These additional data show that the quadrature rule is more computationally efficient than Monte Carlo simulation at accurately determining the AEP.

- We included a note in the previously mentioned paragraph that the advantage of the SGD formulation is that it allows for random samples in each iteration, and the average error is zero.

8) - *The plots on different learning rates, and the discussion on how hyperparameters were selected are appreciated.*

- We have added a dedicated sensitivity analysis to the results.

**References**

[1] Ryan N King, Katherine Dykes, Peter Graf, and Peter E Hamlington. Optimization of wind plant layouts using an adjoint approach. *Wind Energy Science*, 2(1):115–131, 2017.

[2] Jeffery Allen, Ryan King, and Garrett Barter. Wind farm simulation and layout optimization in complex terrain. In *Journal of Physics: Conference Series*, volume 1452, page 012066. IOP Publishing, 2020.

[3] Neil Wu, Gaetan Kenway, Charles A Mader, John Jasa, and Joaquim RRA Martins. py-optsparse: A python framework for large-scale constrained nonlinear optimization of sparse systems. *Journal of Open Source Software*, 5(54):2564, 2020.

[4] Can Zhang, Stephan C Kramer, Athanasios Angeloudis, Jisheng Zhang, Xiangfeng Lin, and Matthew D Piggott. Improving tidal turbine array performance through the optimisation of layout and yaw angles. *International Marine Energy Journal*, 5(3):273–280, 2022.

[5] Ti Zilong and Deng Xiao Wei. Layout optimization of offshore wind farm considering spatially inhomogeneous wave loads. *Applied Energy*, 306:117947, 2022.

[6] Konstanze Kölle, Tuhfe Göçmen, Irene Eguinoa, Leonardo Andrés Alcayaga Román, Maria Aparicio-Sanchez, Ju Feng, Johan Meyers, Vasilis Pettas, and Ishaan Sood. Farmconners market showcase results: wind farm flow control considering electricity prices. *Wind Energy Science*, 7(6):2181–2200, 2022.

[7] Caitlyn E Clark, Garrett Barter, Kelsey Shaler, and Bryony DuPont. Reliability-based layout optimization in offshore wind energy systems. *Wind Energy*, 25(1):125–148, 2022.

[8] Eric Simley, Dev Millstein, Seongeun Jeong, and Paul Fleming. The value of wake steering wind farm control in us energy markets. *Wind Energy Science Discussions*, pages 1–26, 2023.

[9] Rémi Lam, Matthias Poloczek, Peter Frazier, and Karen E Willcox. Advances in bayesian optimization with applications in aerospace engineering. In *2018 AIAA Non-Deterministic Approaches Conference*, page 1656, 2018.

[10] Jichao Li and Mengqi Zhang. Data-based approach for wing shape design optimization. *Aerospace Science and Technology*, 112:106639, 2021.

[11] Paul A Fleming, Andrew PJ Stanley, Christopher J Bay, Jennifer King, Eric Simley, Bart M Doekemeijer, and Rafael Mudafort. Serial-refine method for fast wake-steering yaw optimization. In *Journal of Physics: Conference Series*, volume 2265, page 032109. IOP Publishing, 2022.

[12] Riccardo Riva, Jaime Liew, Mikkel Friis-Møller, Nikolay Dimitrov, Emre Barlas, Pierre-Elouan Réthoré, and Arvydas Beržonskis. Wind farm layout optimization with load constraints using surrogate modelling. In *Journal of Physics: Conference Series*, volume 1618, page 042035. IOP Publishing, 2020.

[13] Andreas Wolf Ciavarra, Rafael Valotta Rodrigues, Katherine Dykes, and Pierre-Elouan Réthoré. Wind farm optimization with multiple hub heights using gradient-based methods. In *Journal of Physics: Conference Series*, volume 2265, page 022012. IOP Publishing, 2022.

[14] J. Criado Risco, R. Valotta Rodrigues, M. Friis-Møller, J. Quick, M. Mølgaard Pedersen, and P.-E. Réthoré. Gradient-based wind farm layout optimization with inclusion and exclusion zones. *Wind Energy Science Discussions*, 2023:1–24, 2023.

[15] José F Herbert-Acero, Oliver Probst, Pierre-Elouan Réthoré, Gunner Chr Larsen, and Krystel K Castillo-Villar. A review of methodological approaches for the design and optimization of wind farms. *Energies*, 7(11):6930–7016, 2014.

[16] David Guirguis, David A Romero, and Cristina H Amon. Toward efficient optimization of wind farm layouts: Utilizing exact gradient information. *Applied energy*, 179:110–123, 2016.

[17] Peter Graf, Katherine Dykes, George Scott, Jason Fields, Monte Lunacek, Julian Quick, and Pierre-Elouan Rethore. Wind farm turbine type and placement optimization. In *Journal of Physics: Conference Series*, volume 753, page 062004. IOP Publishing, 2016.

[18] Pieter Gebraad, Jared J Thomas, Andrew Ning, Paul Fleming, and Katherine Dykes. Maximization of the annual energy production of wind power plants by optimization of layout and yaw-based wake control. *Wind Energy*, 20(1):97–107, 2017.

[19] Nicholas F Baker, Andrew P Stanley, Jared J Thomas, Andrew Ning, and Katherine Dykes. Best practices for wake model and optimization algorithm selection in wind farm layout optimization. In *AIAA Scitech 2019 forum*, page 0540, 2019.

[20] Andrew PJ Stanley, Owen Roberts, Jennifer King, and Christopher J Bay. Objective and algorithm considerations when optimizing the number and placement of turbines in a wind power plant. *Wind Energy Science*, 6(5):1143–1167, 2021.

[21] Carsten Croonenbroeck and David Hennecke. A comparison of optimizers in a unified standard for optimization on wind farm layout optimization. *Energy*, 216:119244, 2021.

[22] Michele Samorani. The wind farm layout optimization problem. *Handbook of wind power systems*, pages 21–38, 2013.

[23] Andrew Ning., Katherine Dykes., and Julian Quick. *Systems engineering and optimization of wind turbines and power plants*, pages 235–292. Wind Energy Modeling and Simulation - Volume 2: Turbine and System. Institution of Engineering and Technology, 2020.

---

## Author Comment (AC3)

**Response to Referee 3**

We greatly appreciate the time taken by the referee to read our manuscript. We have taken into consideration and addressed all comments, questions, and suggestions from the reviewer, and we feel that the revised manuscript is now substantially stronger as a result. Changes made to the text at the request of the reviewer have been highlighted in blue in the revised manuscript. In the following, reviewer comments are repeated in italics and our responses are provided in the bulleted sections of text.

**General comments**

*This manuscript proposes the application of the stochastic gradient descent (SGD) method from deep learning to the wind farm optimization problem, which maximizes annual energy production (AEP) with respect to wind turbine location. It is claimed that this work is the first such application, and it shows results demonstrating significant reduction in computation time for a small set of test cases. The manuscript is generally well-written, and the writing is concise.*

**Specific comments**

*Results are presented for three wind farms of different sizes, with 20 optimizations each with randomized initial designs. The authors are also asked to consider different wind farm shapes. Since there are some tuning parameters in the algorithm (and some tuning was required to achieve favorable results), a concern is whether significant problem-specific tuning is required. Demonstrating consistent results with different wind farm shapes (beyond just the one rectangular domain) would be very compelling.*

- We added material to the manuscript showing the results of circular wind farm layout optimization in addition to examining square wind farms. These changes are reflected in the Methodology on P3L70-80, P5L117, and P8L181, the Application on P8L194-197, as well as a dedicated "circular wind farms" section in the Results.

- As noted in the text on P13L283-P14287, wind farms with circular boundaries yields similar trends in the optimization results to the wind farms with square boundaries.

*There is clearly a cost to the reduction in computation time since constraints are not exactly satisfied. Could the authors comment on the degree of constraint violation and its significance, considering a range of different problems? Perhaps it would be helpful to visualize a Pareto front of computation time versus constraint violation, varying $\eta_T$.*

- We have added a Sensitivity Analysis section to the results to have a dedicated space to answer these and related questions.

- We analyzed the final constraint violations associated with different initial and final learning rates. The analysis indicated that $\eta_T$ is strongly related to the total constraint violation, as illustrated in the left panel of Figure 11 and the accompanying text in P17L323-P18L329.

- We also performed an analysis of the trade-offs between AEP and constraint satisfaction by changing the initial and final learning rates. We include this in the Sensitivity analysis section, in the right panel of Figure 11, as well as with corresponding discussion on P17L329-333.

- Additionally, we included material showcasing an "early stopping" capability of the optimizer, where AEP computations are neglected near the end of the optimization algorithm. This often

yields solutions that exactly satisfy the imposed constraints. The results are shown in Figure 8 and the accompanying discussion is on P16L294-305. For completeness, this "early stopping" option was explicitly added to Algorithm 1.

**Technical corrections**

1) *Line 28-29: Please rephrase - sentence is confusing*

- We changed this sentence to enhance the clarity (P2L29-30):
  The algorithm samples the gradient of a stochastic objective, following the mean gradient by a specified distance, then repeating the process, which amounts to optimizing the expected value of the objective.

2) *Line 44 and onwards: "SciPy SLSQP" should be changed to "SLSQP" - SciPy is a just a package that wraps an implementation of SLSQP*

- We have removed the association of Scipy with the SLSQP algorithm from this point and onward.

3) *Line 77: for completeness, it should be stated at this point where the 8760 comes from, or at least mention that the units are hours*

- We have added a note that this is the number of hours in a year on P4L84-85.

4) *Line 107: I suggest separating the method from the implementation. For example, here, the choice to use the finite-difference method and algorithmic differentiation are aspects specific to the implementation, not aspects of the general method*

- We have moved this information about gradient computation to the Application Section on P9L217-218.

5) *Line 128: "intial" - typo*

- We corrected the typo.

6) *Figure 2: why do these curves have these shapes? Please comment in the manuscript on the trends and whether they agree with your intuition*

- We did this analysis again using the TOPFARM and supercomputer setup associated with the revised results (Figure 3).

- We note that the computational cost grows logorithmically with the number of wind turbines and that there is a more complex scaling associated with increasing the number of samples, which may have to do with the way memory slows down the computer (P10L221-230).

7) *Line 200: again, please comment on \*why\* the gains with SGD are higher with wind farms with more wind turbines*

- In the original manuscript, SLSQP tended to reach the maximum number of iterations as the number of turbines increased. This is reflected in Figure 3 of the originally submitted manuscript, where the deterministic results tend to end at different times when examining small wind farms and the same times when examining large wind farms. So, it's not exactly that SGD did better as the number of wind turbines increased, but rather that SLSQP did worse.

- In our revised results, SGD consistently produced 0.3-0.5% more AEP than the SLSQP counterpart, with the vast majority of results being closer to 0.5% improvement. We discuss this on P13L273-275, remarking that this is likely because the SGD algorithm is able to better explore the design space by initially relaxing the constraints, allowing for some initial constraint violations.

- We have revised the manuscript to reflect that SLSQP generally produces slightly less AEP than SGD, regardless of the wind farm size, and that SGD becomes dramatically less time-consuming to run as the number of turbines is increased (P11L249-P12261 and P13L278-P14291).

8) *Line 214: "constrain" - typo*

- We corrected this typo and ensured this does not happen elsewhere in the text.